# Setting limits: Ethical thresholds to the CEO-worker pay gap

**Carmen Cervone** [1] *, **Andrea Scatolon**[1], **Michela Lenzi**[1], **Anne Maass**[2]

1 Department of Developmental Psychology and Socialization, University of Padua, Padua, Italy,
2 Department of Psychology, New York University Abu Dhabi, Abu Dhabi, United Arab Emirates

* carmen.cervone@unipd.it

**Data Availability Statement:** All study data, materials, and appendices files are available from the OSF database (https://osf.io/9uk2t/?view_only=943ab64c6cf84ec883db93248f34c2f1).

## Abstract

In the discussion about wage inequality, principles of fairness and need for incentives are juxtaposed as opposing motivations for wage inequality acceptance. While previous literature focused on ideal inequality, in two correlational and one preregistered experimental study ($N_{total}$ = 664) we tested the hypothesis of a threshold of inequality acceptance. Participants were asked to indicate what a Chief Executive Officer should earn, ideally (i.e., ideal pay gap) and at maximum (i.e., highest acceptable pay gap), given the wage of a worker. Results showed that individuals generally indicated higher values for highest acceptable than for ideal pay gaps. Study 2 also provides evidence for a minimum degree of inequality that is deemed necessary. Additionally, across studies men preferred and tolerated greater wage inequality than women. In conclusion, these studies provide evidence for a threshold of inequality acceptance and pave the way for new research on the cognitive and motivational underpinnings of attitudes towards economic inequalities.

## Introduction

In the discussion about economic inequality, it is often argued that the concentration of wealth at the top of the social ladder is an essential incentive that encourages individuals to strive for success and make an extra effort to reach higher positions in society. At the same time, however, people universally hold principles of equity and fairness at high value: since childhood, we are willing to reject distributions that are unfair either to ourselves or to others, even at our own cost [1]. Research has shown that a sense of fairness is linked to the willingness to reduce inequality [2], and that the belief that the economic system is illegitimate (hence, unfair) leads to higher demand for the redistribution of resources [3]. Whether inequality is considered an incentive or unfair is particularly relevant when discussing CEO (Chief Executive Officer)-Worker Pay Gap Ratios (henceforth, "PGRs"), as this debate holds tangible consequences on organizational policies and individual wages. Organizational models rely on two competing theories, i.e., the tournament theory [4] and the equity and fairness theory [5, 6]. The tournament model posits that high wage dispersion leads to greater efficiency and productivity in organizations, as workers are motivated to reach higher positions [7]. Consistent with this theory, greater differences in pay were shown to be positively linked to performance [8, 9]; note

**Funding:** This research was supported by grant PRIN 2017 #2017924L2B of the Italian Ministry of Education, University and Research (MIUR), entitled "The psychology of economic inequality", to the last author. The funders had no role in study design, data collection and analysis, decision to publish, or preparation of the manuscript.

**Competing interests:** The authors have declared that no competing interests exist.

however that the evidence is mixed [10]. On the contrary, the equity model states that high wage inequality leads to decreased productivity, as workers perceive the organization as unfair and are less motivated to make an effort. Consistent with this notion, greater pay differences are negatively correlated with employees' negative perception of the company [11], diminished perception of leaders' efficacy and charisma [12], lower integration and poorer performance of top executives [13], and higher likelihood of fraud [14], as well as reduced purchasing intention [15] and a negative opinion of the company [11] by consumers.

The tournament theory and the equity and fairness theory have often been juxtaposed by organizational research, but, to our knowledge, rarely combined. We suggest, instead, that the coexistence of these two notions may be explained by the belief that inequality is good, but only as long as it falls within a certain range. Even 4-year-old children are ready to reward merit, thus creating some degree of inequality, yet they refuse to cross a certain inequality threshold [16]. In other words, inequality acceptance might be imagined as a curve in which the turning point is represented by one's threshold of inequality tolerance, the "ethical ceiling" mentioned by Osberg and Smeeding [17]. This paper aims to explore this threshold in the context of PGRs, and to investigate whether this threshold is affected by gender, as will be outlined below.

## A threshold to wage inequality: Ideal vs. Highest acceptable pay gap

Research consistently shows that individuals prefer wealth distributions that are more equal than the current ones but that still show some degree of inequality [17–19]. Even for pay gaps, across the world people seem to agree that a certain level of inequality is ideal, although large cross-cultural differences in the "amount of inequality" exist [17, 20, 21]. To our knowledge research on economic inequality and pay-gap ratios has yet to explore whether there is a limit to the inequality that individuals deem ethically acceptable, namely the highest acceptable pay gap. Osberg and Smeeding [17], for example, conceptualise the "ethical ceiling" as the ideal PGR, with the assumption that ideal ratios are the threshold for acceptable inequality. We argue instead that these are two different concepts, as individuals have a tolerance for inequality that goes beyond their preference. Hence, the primary, overarching aim of this research was to investigate whether and where people would set the limits for CEO-worker pay gaps.

The idea of a maximum acceptable wealth or pay gap is by no means new. Plato, in Book 5 of The Laws, discusses the ethical underpinnings of wealth distribution, making very precise policy recommendations for a hypothetical city state. To avoid hatred and divisions among individuals, he suggests prohibiting excessive accumulation of wealth and to distribute land and housing across four social classes. In his ideal society, each man is guaranteed the possession of one lot (the minimum) whereas the maximum is set to four times this amount: "Let the limit of poverty be the value of a lot [. . .] This the legislator will permit a man to acquire double or triple, or as much as four times the amount of this" [22].

In advanced social democracies, the maximum pay gap is generally defined by two interrelated rules, concerning minimum wage and executive salaries, respectively. According to Bruni [23], 11 EU member states have implemented a binding cap policy on remuneration in the public and/or semipublic sectors, whereas 17 states have performance-related pay regulations. From an applied point of view, since laws generally define thresholds and since these regulations are already present in some (albeit few) countries, maximum and minimum wage thresholds may be envisaged as more realistic than ideal or suggested ones, which can hardly be transformed into a tangible policy. Therefore, exploring the individual and contextual determinants of people's threshold for wage inequality might prove more consequential for the political and social landscape than individual preferences or ideals.

Although we do not explore this issue here, the highest acceptable PGR may also be interesting from a psychological point of view. Since the threshold can be considered a personal norm, one may predict that a violation of this norm (i.e., higher inequality) may be perceived as more illegitimate than the violation of one's ideal, thus possibly leading to greater anger and willingness to engage in collective action to reduce inequality. This may be especially true when inequality beyond the threshold is perceived as unethical, i.e., as a moral violation. Thus, there are both theoretical and applied reasons for separating highest acceptable from ideal PGRs that point to the relevance of investigating them as separate constructs.

## Gender differences in preferences for pay ratios

Like other disadvantaged groups, women have often been found to endorse politically progressive values and policies. This ideological gender gap is a relatively modern phenomenon, having originated around the 80s; although cross-cultural differences in historical trends are present, the ideological gender gap was consistently shown in European and Northern American countries [24]. Among others, women are more environmentally aware, are less likely to support force and military interventions, and are more likely to favor gun control [25]. They also tend to hold more equalitarian economic views: compared to men, women are more supportive of government interventions, redistribution, social spending, and welfare programs in favor of the less fortunate [26–28]. Importantly, gender differences are well documented also with respect to values and beliefs known to be related to redistribution and social justice attitudes. In particular, women have consistently been found to be lower in social dominance orientation (SDO) [29] and, hence, more opposed to hierarchies in which some groups dominate over others. They also tend to be less prone to competition, attribute less positive outcomes to competition [30] and believe less strongly in gender-specific system justification [31].

When it comes to wage inequality, this is also the case. According to the 2023 Global Gender Gap Report by the World Economic Forum, the gender gap is still a global issue, with women being disadvantaged in economic participation and opportunities compared to men across all reviewed countries. For example, in the United States, women currently earn 99¢ for every 1$ earned by men, even when controlling for qualifications and job characteristics [32]. In line with such disadvantage, women tend to indicate lower ideal PGRs, namely they prefer somewhat smaller pay gap ratios, compared to men [33, 34]. These preferences also appear to have a real-word impact: companies led by female CEOs also tend to be more egalitarian, as the CEO wage (but not the worker wage) tends to be lower [35]. Put together, this evidence suggests that women may be less tolerant of CEO-worker pay gaps and, possibly, more favorable of smaller caps on executive remuneration compared to men. At the same time, however, there is also evidence pointing to the fact that men and women deem set wages as similarly fair or unfair [36], which would suggest that gender may not affect concerns about wage equality. Therefore, the second aim of this research was to investigate whether gender differences are also present in wage inequality thresholds.

## Legitimizing PGRs: Wage-setting criteria

There are several factors that may affect the levels of inequality that individuals prefer and are willing to accept. For example, people who see themselves as wealthier tend to indicate higher pay gaps between CEOs and workers as ideal [37]. Here, we explore the legitimization of wage differentials through wage-setting criteria.

To date, relatively little is known about the criteria that lay people believe determine wages. Evans and colleagues [38] consider family need, performance, and authority as criteria for wage determination, which may legitimize differences in pay. The authors define family need

as the need to support a family and children with one's own wage, performance as how hard one works and how good one is at the job, and authority as number of years spent in education as well as authority over others. They find cross-cultural consensus on performance as justifying high pays, whereas authority appears to be particularly relevant in some countries, but not in others. Kiatpongsan and Norton [20], instead, consider responsibility, performance, and effort, and find that individuals with different beliefs indicate ideal PGRs that are smaller than perceived ones, though it remains unclear whether and to which degree these criteria predict perceived or ideal PGRs. However, many criteria that determine wages in real life (such as shifts, or skillset needed) are not considered in previous studies. Consistent with research on fairness and meritocracy, people believe that a fair distribution of resources ought to take differences in effort into account [39]. Consequently, one may assume that fair wages–and hence, fair PGRs–should reflect differences in effort, skillsets or stress levels that come with different roles within an organization. In other words, the relative importance that people attribute to different criteria should be predictive of their ideal and highest acceptable PGRs.

### Hypotheses

Previous literature has evidenced how wage inequality can simultaneously have positive and negative outcomes on organizations, as a high inequality can provide an incentive for productivity, but also negatively affect employees, management, and the public image of the organization. We argue here that this apparent contradiction is justified by the idea that inequality is preferable–but only within a certain, individual, threshold. Therefore, the main aim of this research is to investigate whether people perceive a threshold of wage inequality that should not be crossed, since it creates inequalities that can not be accepted (regardless the specific features of the work positions). To do so, we asked participants how much a CEO should earn ideally and at maximum, based on the wage of the lowest-paid worker in the same company. We predicted that the highest acceptable wages indicated by participants would exceed ideal ones (Hp1).

Furthermore, given that women (a) hold more egalitarian values, and (b) prefer lower income inequality in societies compared to men, we expected women to desire lower PGRs (Hp 2a). In Study 2 we also tested whether these gender differences be mediated by the differential endorsement of inequality-related ideologies such as SDO, social mobility beliefs and merit beliefs (Hp 2b).

Finally, to explore potential mechanisms underlying ideal and highest acceptable PGRs, the second aim was to investigate the criteria that, in participants' minds, guide wage setting. We predicted that people who primarily value those criteria that are typically associated with CEOs (e.g., competence), but not those typical of the worker (e.g., fatigue) would also accept and desire higher PGRs (Hp3).

**Open practices and transparency.** All study data, materials, and appendices are accessible at this link: https://osf.io/9uk2t/?view_only=943ab64c6cf84ec883db93248f34c2f1. We report how we determined our sample size, all data exclusions, all manipulations, and all measures in the studies.

## Study 1

We predicted that highest acceptable PGRs would be higher than ideal PGRs, consistently with the results of a Pilot Study (Appendix A). Additionally, we explored which criteria participants believe (a) are applied and (b) ought to be applied in wage setting and tested whether these predicted PGRs.

## Method

**Participants.** Data collection was run between 29th January and 7th February 2020. In total, 334 participants accessed the questionnaire; of these, 12 did not complete it and were excluded. Thus, 322 participants completed the questionnaire (200 women, 117 men, 5 non-binary; $M_{age}$ = 25.61; $SD_{age}$ = 10.36), of whom 47% students; they were left leaning ($M$ = 43.98, $SD$ = 25.31) and saw themselves as better off than the average Italian family ($M$ = 54.36, $SD$ = 15.34). The results of a post-hoc sensitivity power analysis ran on G*Power 3 [40] showed that our sample had 80% statistical power to detect at least an effect size of $f$ = .15 for the interaction between our predictors.

**Procedure.** As part of a research lab, four groups of first-year Psychology students attending a Social Psychology course handled data collection during the first semester. As they did so independently, they added variables of their interest to the questionnaire; nonetheless the variables included in this study were unaffected by other variables.

Participants were first asked to indicate in two open text-boxes how much a CEO should earn ideally and at most compared to the least-paid worker of the same company (i.e., 1000€). A subgroup of participants ($N$ = 131) was also asked to indicate to what degree certain criteria are currently used (perceived criteria) and ought to be used (ideal criteria) to determine wages, using a 7-point Likert scale. Data on this variable is only available for a subgroup of participants since two of the groups who handled data collection decided not to include wage-setting criteria in the questionnaire.

Four items assessed job-related fatigue and included total amount of hours, distribution of hours in shifts, whether the job was physically exhausting, and whether it was emotionally exhausting ($\alpha_{perceived}$ = .69; $\alpha_{ideal}$ = .70). Two items assessed competence, namely required skills and responsibility required by the role ($\alpha_{perceived}$ = .79; $\alpha_{ideal}$ = .67). Three items assessed utility, namely money circulated by the activity, benefits produced for the company, and benefits produced for society ($\alpha_{perceived}$ = .80; $\alpha_{ideal}$ = .64). Participants could also indicate additional criteria through a text box.

Finally, participants were asked demographic information: gender, age, education, region of birth and residence, family income, number of family members, SSES (i.e., "Compared to the average Italian family, my family is. . .", from 0—*worse off* to 100—*better off*), political orientation (from 0—*left-wing* to 100 –*right-wing*) and religiosity; and they were debriefed.

Studies were approved by Institutional Review Board of the University of Padua (Protocol Number 3352). All studies required participants to sign a written informed consent in which they were informed about their rights as participants and details about data treatment.

## Results

**Ideal and highest acceptable pay-gaps.** Ideal PGR ranged from 1 to 150 ($M$ = 6.49; $SD$ = 14.69) and highest acceptable PGR from 1 to 1000 ($M$ = 18.78; $SD$ = 98.08). Most participants ($N$ = 183, 57%) indicated a higher value for highest acceptable than for ideal PGR, while 115 (36%) indicated the same amount for both variables, and only 24 (7%) indicated lower highest acceptable PGRs than ideal ones. Analyses ran after exclusions of the latter group of participants are available in Appendix H of the supplementary materials.

As participants were able to indicate any value above 1000€, we divided wages by 1000 and tested the difference between PGRs through both linear and non-parametric methods. As for the linear method, our data required two additional steps before analysis: first, we excluded outliers on any of the two variables through the interquartile-range (IQR) approach for detecting extreme outliers—i.e., three IQRs above the third quartile [41], then values were adjusted

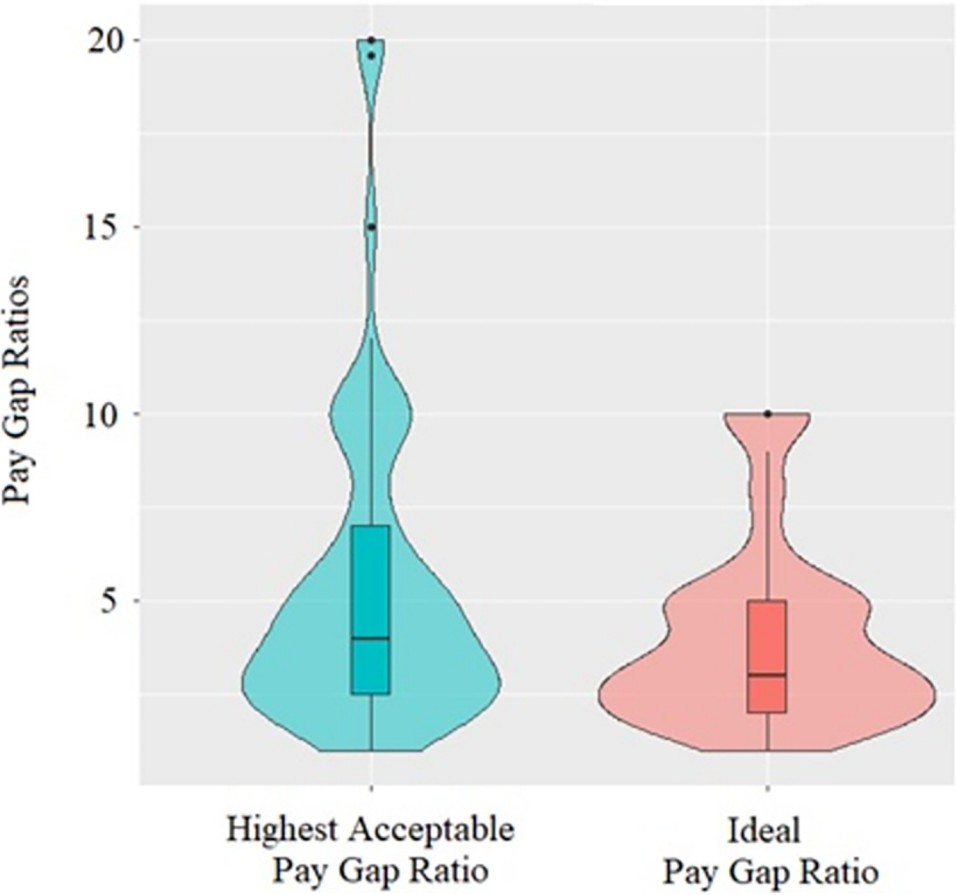

**Fig 1. Ideal and highest acceptable PGRs (Study 1).**

through log-linear transformation to correct for skewedness. Across studies, outliers are excluded for linear analyses involving PGRs, while all other analyses employ the full sample.

A Wilcoxon Signed-Ranks test indicated that highest acceptable PGR was greater than ideal PGR, $Z = -10.01$, $p < .001$, $\eta^2 = .31$, which was confirmed by a paired-samples t-test, $t(301) = -9.57$, $p < .001$, $d = -.55$ after outlier exclusion ($N = 20$; Fig 1).

**Wage-setting criteria.** To test whether perceived and ideal criteria were perceived differently, we ran a 2(perceived vs ideal) x 3(type of criteria) ANOVA with repeated measures on both variables, using a Greenhouse-Geisser correction. We found a main effect of perceived vs ideal criteria, $F(1, 134) = 204.06$, $p < .001$, $\eta_p^2 = .60$, with participants believing that criteria should ideally have greater weight in determining wage, and a main effect of type of criteria, $F(2, 261) = 78.08$, $p < .001$, $\eta_p^2 = .37$, showing that competence was believed to be more important than the other two criteria. The interaction between the two, $F(2, 264) = 6.85$, $p = .001$, $\eta_p^2 = .05$, showed that, according to our participants, all three dimensions should have greater weight in determining salaries, but this was particularly true for fatigue, $t(134) = -14.95$, $p < .001$, followed by competence, $t(134) = -12.05$, $p < .001$, but less so for utility, $t(134) = -8.58$, $p < .001$. Then, we ran four regression models with PGRs as outcomes and ideal or perceived fatigue, competence, and utility as predictors, but found no effects ($ps > .10$).

## Discussion

Study 1 confirmed our hypothesis that ideal pay gaps and highest acceptable pay gaps are two distinct concepts in the majority of cases, with individuals indicating highest acceptable PGRs greater than ideal PGRs. Moreover, our results show that men prefer and tolerate higher wage inequality compared to women. There are several potential psychological processes that may lead to this outcome: for example, men may believe more strongly in system-justifying ideologies, such as meritocracy or social mobility [31], or they may have a stronger need for well-defined hierarchies [42], and thus accept greater distance between the low-status employee (the worker) and the high-status employee (the CEO). Hence, we tested the role of these potential mediators in Study 2.

As for wage setting criteria, our results showed that while participants distinguished between perceived and ideal criteria, criteria themselves did not determine PGRs. One reason for this may be that in Study 1, criteria were not pretested: therefore, our a-priori selection of criteria as more relevant for low-wage workers or CEOs may have not matched with participants' perceptions (e.g., fatigue could be associated with the physical work of factory workers, but also with the long working hours of managers). We solved this issue in Study 2 with a new, pre-tested measure, that allowed us to reliably separate between criteria that are typical of workers vs. CEOs.

## Study 2

The first aim of this study was to understand the processes behind the gender differences observed in Study 1. Specifically, we hypothesized that SDO, social mobility beliefs and ideologies of merit would mediate the relation between gender and PGR.

As our second aim, we wanted to investigate a potential lowest acceptable PGR, i.e., the minimum possible wage participants attributed to the CEO, in order to test whether individuals also have a threshold of the inequality they feel is necessary. We predicted that highest acceptable PGR would exceed ideal PGR, which in turn would exceed lowest acceptable PGR. Since ideal PGR responses could possibly have functioned as anchor for highest acceptable PGRs in our prior studies, we also reversed the order of the measures in this study, so that limits (upper and lower bounds) to pay gaps preceded ideal pay gaps.

Finally, we attempted to investigate wage setting criteria in a more systematic way, employing a new, pre-tested set of criteria. Specifically, we predicted that participants who attributed more importance to the criteria typical of the CEO (e.g. "leadership ability") would also prefer and accept higher PGRs.

## Method

**Participants.** Data collection was run between 23$^{rd}$ April and 8$^{th}$ May 2020. Students of a Political Psychology course were asked to complete the questionnaire and send it to five other people. In total, 303 people accessed the questionnaire; of these, 3 did not give their consent to data processing and 54 did not complete the questionnaire. Thus, the sample consisted of 246 participants, including 159 women and 87 men. Work status was unbalanced across genders, $\chi^2(1, N = 246) = 6.45$, $p = .011$, with 69% of women being students compared to 53% of men, creating a confound between gender and work status. Thus, we ran the analyses on a sub-sample balanced for work-status, by randomly excluding female students. Analyses on the full sample are reported in Appendix B of the Online Supplementary Materials.

The sub-sample consisted of 192 participants, 87 men and 105 women ($M_{age} = 28.94$, $SD_{age} = 12.21$) of comparable work status, $\chi^2(1, N = 192) = .004$, $p = .949$. The sample was politically left-leaning ($M = 32.66$, $SD = 23.24$), one-sample $t(191) = -10.34$, $p < .001$, and the SSES

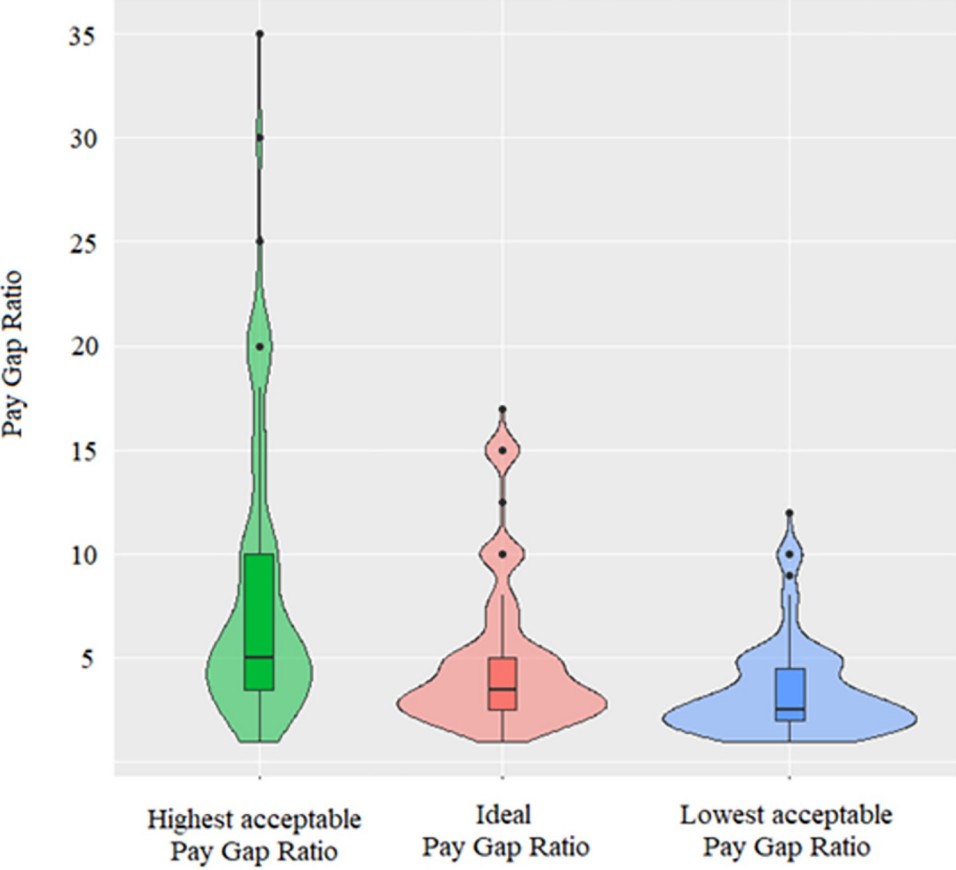

**Fig 2. Highest acceptable, ideal and lowest acceptable PGRs in Study 2.**

compared to the national average exceeded the midpoint ($M = 57.39$, $SD = 15.20$), one-sample $t(191) = 6.73$, $p < .001$. The results of a post-hoc sensitivity power analysis showed that our sample had 80% statistical power to detect at least an effect size of $f = .19$ for the interaction between our predictors.

**Procedure.** *Ideal, highest acceptable and lowest acceptable pay gaps.* To facilitate comprehension, we presented participants with the following variation of the PGR measure: "You will now be shown the descriptions of two people working for the same company. It's an Italian company that produces food and has 200 employees. Francesco S. is a factory worker for this company and his is the job with the lowest wage within the company. He earns 1000€ (post-tax) per month. Alessandro B. is the CEO of the same company (the employee with the highest managerial role in the company) and his is the job with the highest wage within the company.".

To make the task easier, participants were also shown a figure presenting 23 pay gaps (going from 1:1 to 55:1), so that they could picture the ratios (see Fig 2 in Appendix C of the Online Supplementary Materials). Participants were then asked to indicate a ceiling for the wage of the CEO, considering that the worker earned 1000€ per month ("ceiling" was defined as the highest monthly salary, beyond which the wage would become excessively high and thus unacceptable). In the same way, they were asked to indicate a minimum level for the wage of the CEO, i.e., the minimum monthly salary, below which the wage would become excessively low and thus unacceptable. Finally, they were told that since in reality there is a certain

variability in wage differentials, we were asking them to envisage a company that according to them was ideal, and to indicate what they would want the wage of the CEO to be. Once again, participants provided their responses in an open text box.

*Need for legislation and relative usefulness of ideal vs. Maximum pay gap rules*. Participants were asked if they thought there ought to be a law that sets a ceiling to the wage of the CEO. Answers were provided on a slider going from 0 (*absolutely not*) to 100 (*absolutely yes*). Subsequently, participants were asked "from a practical point of view, do you think it's more important to (a) define an ideal wage gap that companies should approach as closely as possible, or (b) define a maximum threshold that companies must not exceed.". Participants answered on a slider going from 0 (*ideal gap*) to 100 (*maximum threshold*).

*Wage setting criteria*. A new pre-tested measure of wage-setting criteria (Appendix D) distinguished, also according to an exploratory factor analysis, between criteria associated with low-status workers (i.e., physical exhaustion and shift work, Spearman-Brown = .73) and criteria associated with CEOs (competence, responsibility, leadership ability, organizational ability, and knowledge of languages; $\alpha$ = .78). Participants were asked to rate how much each criterion was important in setting the wage of a person (from 1 –*not at all* to 7 –*extremely*).

*Ideology*: *Social dominance, merit and social mobility*. To test whether our ideology variables mediated the relation between gender and PGR, we employed the short version of the SDO7 scale (Ho et al., 2015), while social mobility beliefs and meritocracy beliefs were assessed through scales by Day and Fiske (2017). Answers were provided on a 7-point Likert scale going from *disagree* to *agree* (SDO: $\alpha$ = .75; social mobility: $\alpha$ = .73; meritocracy: $\alpha$ = .81). Furthermore, participants were asked if wages should be more based on need or merit. Answers were provided on a slider going from 0 (*need*) to 100 (*merit*).

*Demographic variables*. Finally, participants were asked to indicate gender, age, education level, work status, political orientation (three items, answers were provided on a slider going from 0 –*left* to 100 –*right*; $\alpha$ = .93), SSES and annual family income. Finally, they were debriefed.

## Results

Ideal PGR ranged from 1 to 100, highest acceptable PGR from 1 to 200, and lowest acceptable PGR from 1 to 30. Again, most participants ($N$ = 168, 87%) indicated higher values for highest acceptable PGR than ideal PGR, while 22 (12%) indicated equal values, and two indicated lower values. Similarly, most participants ($N$ = 162, 84%) indicated higher values for ideal PGR than for lowest acceptable PGR, whereas 21 (11%) indicated equal values, and 9 (5%) indicated lower values. Finally, only one participant indicated lower values for highest acceptable than lowest acceptable PGR and three participants indicated equal values, whereas the rest of the sample indicated higher values. Analyses ran after exclusions of participants who indicated ideal > highest acceptable PGRs, lowest acceptable > ideal PGRs, and lowest acceptable > highest acceptable PGRs are available in Appendix H of the supplementary materials. Wilcoxon Signed-Ranks tests indicated that highest acceptable PGR was greater than ideal PGR, $Z$ = -11.23, $p < .001$, $\eta^2$ = .66, which in turn was greater than lowest acceptable PGR, $Z$ = -10.20, $p < .001$, $\eta^2$ = .54.

After exclusion of outliers ($N$ = 13, 10 men and 3 women; $N_{ideal}$ = 10, $N_{highest}$ = 4, $N_{lowest}$ = 6), ideal PGR ranged from 1 to 17, highest acceptable PGR from 1 to 35, and lowest acceptable PGR from 1 to 12. Distribution of scores is presented in Fig 2.

**Pay gap ratios as a function of gender.** A 3 (type of PGR: lowest acceptable vs. ideal vs. highest acceptable) x 2 (gender) ANOVA with repeated measures (Greenhouse-Geisser correction) on the first variable confirmed the predicted main effect for type of PGR, $F(2, 291)$ =

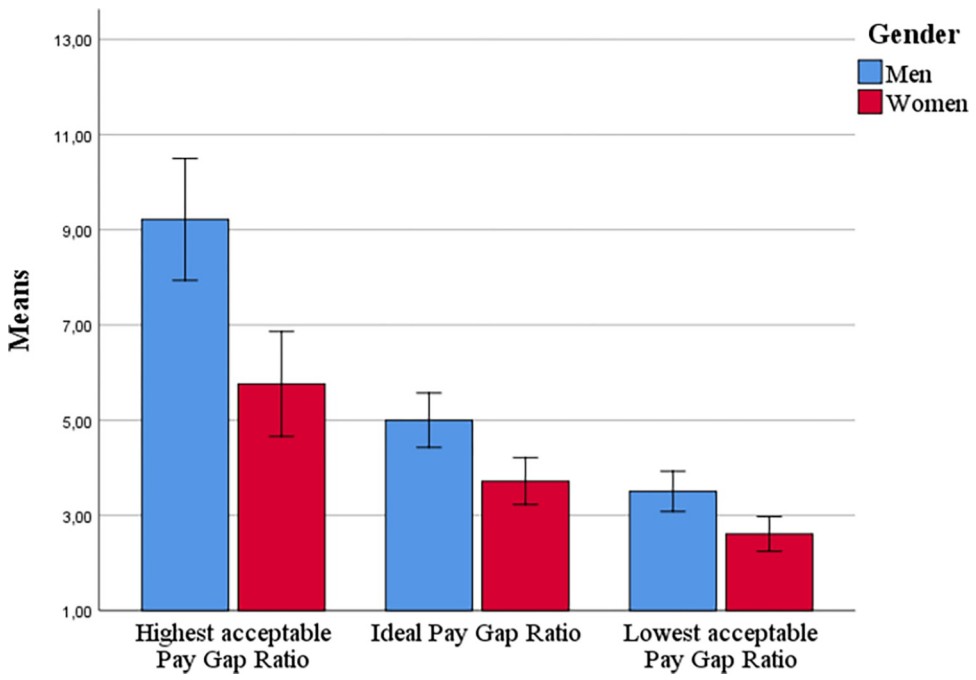

**Fig 3. Ideal, highest acceptable and lowest acceptable PGRs as function of gender (Study 2).** *Note*. Means are presented in raw scores instead of log-transformed values to facilitate interpretation. Error bars: 95% CI.

300.18, $p < .001$, $\eta_p^2 = .63$. Highest acceptable PGR ($M = 8.23$, $SD = 6.68$) exceeded ideal PGR ($M = 4.87$, $SD = 3.52$), $t(178) = 15.69$, $p < .001$, which in turn exceeded lowest acceptable PGR ($M = 3.36$, $SD = 2.21$), $t(178) = 11.79$, $p < .001$.

An additional main effect of gender, $F(1, 177) = 14.78$, $p < .001$, $\eta_p^2 = .08$, revealed that men ($M = 6.67$, $SD = 4.17$) indicated higher PGRs than women ($M = 4.60$, $SD = 3.22$). Furthermore, an interaction between gender and PGR, $F(2, 291) = 5.37$, $p = .008$, $\eta_p^2 = .03$, indicated that the gender difference was strongest for highest acceptable PGR, $t(177) = 4.22$, $p < .001$, intermediate for ideal PGR, $t(177) = 3.53$, $p = .001$, and smallest for lowest acceptable PGR, $t$ (177) = 2.71, $p = .007$ (Fig 3). Furthermore, lowest acceptable PGR (log-transformed) exceeded 0 for both men, one-sample $t(80) = 18.02$, $p < .001$, and women, one-sample $t(104) = 16.19$, $p < .001$, suggesting that both deemed some degree of pay gap between CEOs and workers necessary.

**Gender and ideology.** We had predicted that gender differences in PGRs be, at least in part, explained by differences in ideologies. Initial t-tests while including outliers revealed that men ($M_{SDO} = 2.84$, $SD_{SDO} = 1.09$) were more strongly in favour of hierarchies than women ($M_{SDO} = 2.49$, $SD_{SDO} = .90$), $t(167) = 2.39$, $p = .018$. Also, men ($M_{Merit} = 3.57$, $SD_{Merit} = .75$) endorsed the merit principle more strongly than women did ($M_{Merit} = 3.33$, $SD_{Merit} = .91$), $t$ (190) = 2.01, $p = .046$. However, men and women held similar social mobility beliefs, $t(190) = .25$, $p = .801$, and provided similar responses when need and merit were pitted against each other, $t(190) = .86$, $p = .393$. We will therefore only focus on the two ideologies where gender differences emerged, namely SDO and merit beliefs.

*Mediation models.* To test whether ideology mediated the effect of gender on PGRs, we ran mediation models with the SPSS macro PROCESS, model 4 (Hayes, 2017), with separate models for each mediator. For all mediation analyses, we calculated indirect effects using a percentile bootstrapping procedure with 10.000 resamples.

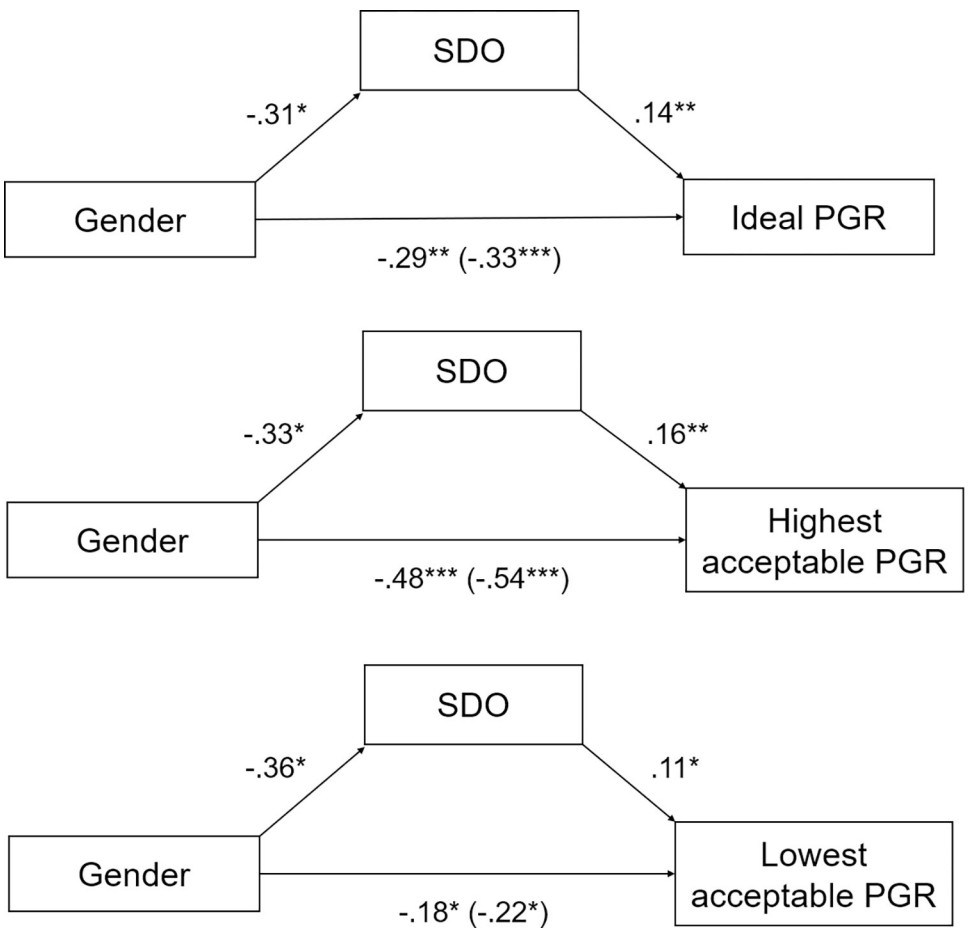

**Fig 4. Mediation models with gender as predictor, SDO as mediator, and PGRs as outcomes (Study 2).**

Regression models with total and direct effects are presented in Appendix E. As for SDO (Fig 4), results of the mediation analyses showed a partial mediation effect for ideal PGR (indirect effect: $B$ = -.04, BootLLCI = -.10, BootULCI = -.003), highest acceptable PGR (indirect effect: $B$ = -.05, BootLLCI = -.13, BootULCI = -.003) and lowest acceptable PGR (indirect effect: $B$ = -.04, BootLLCI = -.10, BootULCI = -.004). It is relevant to note that after exclusions of participants who indicated ideal > highest acceptable PGRs, lowest acceptable > ideal PGRs, and lowest acceptable > highest acceptable PGRs, the mediations are not significant anymore (see Appendix H). As for merit beliefs, results of the mediation analyses showed no mediation effect neither for ideal PGR nor for highest acceptable PGR, nor for lowest acceptable.

*Additional predictors.* As for the other predictors, mobility beliefs were positively linked to ideal PGR, $r(182)$ = .16, $p$ = .036, but not to highest acceptable PGR, $r(188)$ = .14, $p$ = .064, or lowest acceptable PGR, $r(186)$ = .04, $p$ = .588. The belief that merit should be more relevant than need in wage setting was positively correlated with ideal PGR, $r(182)$ = .17, $p$ = .023, and lowest acceptable PGR, $r(186)$ = .18, $p$ = .016, but not highest acceptable PGR, $r(188)$ = .09, $p$ = .217.

**Wage-setting criteria.** We first tested whether the importance assigned to CEO- and worker-related wage setting criteria predicted ideal, highest acceptable and lowest acceptable PGRs. We ran three regression analyses, one for each type of PGR, using the two wage setting

criteria as predictors. Worker-related wage setting criteria predicted PGRs negatively (ideal PGR: $B$ = -.14, 95% CI [-.24, -.05], $t$ = -2.94, $p$ = .004; highest acceptable PGR: $B$ = -.14, 95% CI [-.27, -.02], $t$ = -2.27, $p$ = .025; lowest acceptable PGR: $B$ = -.10, 95% CI [-.20, -.01], $t$ = -2.21, $p$ = .028), whereas CEO-related wage setting criteria were small and unreliable positive predictors of PGRs (ideal PGR: $B$ = .125, 95% CI [.01, .24], $t$ = 2.11, $p$ = .037; highest acceptable PGR: $B$ = .13, 95% CI [-.03, .28], $t$ = 1.63, $p$ = .105; lowest acceptable PGR: $B$ = .11, 95% CI [-.01, .22], $t$ = 1.85, $p$ = .066).

To investigate whether gender predicted differences in importance attributed to criteria, we ran a 2 (type of criteria: CEO vs. worker) x 2 (gender) ANOVA with repeated measures (Greenhouse-Geisser correction) on the first variable (while including outliers), which revealed a main effect for type of criteria, $F(1, 190)$ = 7.60, $p$ = .006, $\eta_p^2$ = .04, and an interaction between the two variables, $F(1, 190)$ = 9.11, $p$ = .003, $\eta_p^2$ = .05. Overall, criteria linked to CEOs ($M$ = 5.50, $SE$ = .05) were deemed more important in determining wages than those linked to workers ($M$ = 5.27, $SE$ = .07). This effect was entirely due to male participants who judged CEO criteria ($M$ = 5.57, $SE$ = .08) as more important than worker criteria ($M$ = 5.09, $SE$ = .10), $t(86)$ = 3.86, $p$ < .001, whereas women judged CEO ($M$ = 5.44, $SE$ = .07) and worker criteria ($M$ = 5.46, $SE$ = .09) equally important for determining wages, $t(104)$ = -.20, $p$ = .845. Alternatively, men and women valued the criteria linked to CEO's to similar degrees, $t(190)$ = 1.20, $p$ = .233, but women judged the criteria associated with workers as more important for determining wages than men did $t(190)$ = -2.83, $p$ = .005.

At this point, we ran mediation models to test whether perceived importance of worker criteria mediated the gender difference on the PGRs. Regression models with total and direct effects are presented in Appendix E. Results of the mediation analyses showed a partial mediation effect for ideal PGR, as shown in Fig 5 (indirect effect: $B$ = -.04, BootLLCI = -.09, BootULCI = -.003), but not for highest acceptable PGR or lowest acceptable PGR.

**Need for legislation and relative usefulness of ideal vs. Maximum pay gap rules.** Including outliers, the majority of participants (70%) desired a law that puts a cap on executive salaries (i.e., indicated scores higher than the mid-point), binomial < .001. This desire was stronger among women (76%) than among men (63%), $\chi^2(1, N = 186)$ = 3.83, $p$ = .050. Also, participants who desired such law were more left-wing ($M$ = 29.07, $SD$ = 22.86) than those who did not ($M$ = 41.44, $SD$ = 22.50), $t(184)$ = 3.40, $p$ = .001. However, when asked whether it is more useful to define an ideal wage gap that companies should comply to as best as possible or to define an upper limit that companies must not exceed, participants opted more frequently for the former (68%) than for the latter solution, binomial < .001, regardless of participant gender and political opinion.

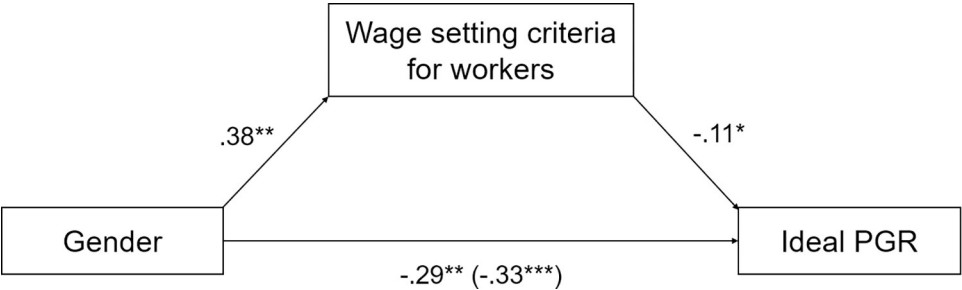

**Fig 5. Mediation models with gender as predictor, worker criteria as mediator, and ideal PGR as outcome (Study 2).**

After excluding outliers, participants who believed that there ought to be a cap on executive salaries, $t(180) = 2.64$, $p = .009$, and those who believed that an upper limit for the wage gap would be more useful, $t(179) = 2.11$, $p = .036$, indicated smaller gaps for highest acceptable PGR, while there was no difference on ideal PGRs and lowest acceptable PGRs, all $p$'s $\geq .14$.

## Discussion

This study confirms our previous results that ideal and highest acceptable pay gaps are two distinct concepts and provides evidence for the lowest acceptable PGR, that is a level of wage inequality people feel is necessary between a CEO and a worker. Furthermore, PGRs were affected by importance attributed to characteristics typical of worker roles: the more participants believed that criteria associated with workers are important in wage setting, the less wage inequality they required, preferred and tolerated.

Secondly, this study confirms the relation between gender and PGRs and provides evidence for two potential psychological processes driving the gender differences, namely SDO and the relevance attributed to worker criteria. Nonetheless, the partial mediation effects that emerged in our study were small and not consistent across the three PGRs: hence, other factors may be explaining the gender differences, as discussed below.

Finally, in Study 2 almost 80% of women and over 60% of men favored a law putting a cap of executive pay; participants who supported such a law deemed lower maximum pay gaps to be ethically acceptable. However, most participants also thought it more useful to define an ideal wage gap that companies ought to comply to rather than to define an upper limit that companies must not exceed. Participants may assume that setting an upper limit would lead the gap to reach the upper limit in most cases, and thus they believe it more useful to define an ideal gap. Unfortunately, the formulation of this item was somewhat ambiguous, making it difficult to draw unequivocal conclusions about the participants' preferences: the item did not specify whether we were referring to a maximum gap or a maximum wage. Thus, participants may have preferred the ideal gap either because they preferred relative gaps over absolute caps, or because they favored "best practice" rules over laws. As such, we decided to solve this issue in Study 3.

## Study 3

The within-participants design of Study 1 might have created a demand effect and led participants to assume that they ought to provide different answers to the two items. Therefore, in Study 2 we employed a between-participants design and once again, we predicted that highest acceptable gaps would be greater than ideal gaps. Additionally, we aimed to explore the predictive value of a new set of wage-setting criteria. Finally, in this study we also explored the motivations that lead participants to select highest acceptable and ideal PGRs. This study is preregistered at: https://aspredicted.org/blind.php?x=pe5qi9.

### Method

**Participants.**   Data collection was run between 27th April and 5th May 2021. G*Power 3 [40] indicated a sample of 138 to detect an effect size of $d = .43$ (based on Pilot Study 2, Appendix C) with $\alpha = .05$ and 80% power. To allow for potential exclusions, additional participants were sampled. 177 participants answered our online survey; of these, 23 participants failed the manipulation check and three failed one attention check, and were therefore removed. Thus, the final sample consisted of 151 Prolific workers (68 women, 82 men, 1 non-binary; $M_{age} = 26.31$; $SD_{age} = 7.30$), of whom 50% students; participants leaned to the left, $t(150) = -10.90$, $p$

< .001 (*M* = 32.50, *SD* = 19.73), and saw their social standing as similar to the average Italian family (*M* = 52.21, *SD* = 15.12).

**Procedure.** Participants were randomly assigned to one of two conditions (highest acceptable PGR vs ideal PGR). To help them answer the PGR item, in this study participants were also provided with a figure visualizing different PGRs from 1:1 to 55:1 (Pilot Study 3—Appendix F). Then, they answered items exploring the motivations that led participants to select highest acceptable and ideal PGRs. Motivations were investigated through item pairs (from 1 – *I did not think about this* to 7 – *This motivation was fundamental in my decision*) assessing equality/fairness (Spearman-Brown = .76), merit (Spearman-Brown = .25), and quality of life (Spearman-Brown = .31; we were unable to use the latter two due to low reliability), and a bipolar item assessing which was more relevant in deciding the wage of the CEO, equality (1 –"maintaining equality and equity between the two workers, without creating unfairness or privileges") or merit (6 –"Rewarding the merit of the CEO, who has greater responsibility and competence"). These specific motivations were selected since they emerged from Pilot Study 3 (Appendix F). Additionally, we included one item that we predicted to be higher in the ideal PGR condition ("I wanted to create a balance between the two categories"), and one that we predicted to be higher in the highest acceptable PGR condition ("I thought there were no justifications for a greater gap").

At this point, we assessed participants' need for legislation (i.e., if they thought there should be a law setting a ceiling to CEO wages, from 0 –*absolutely not* to 100 –*absolutely yes*), and perceived usefulness of ideal vs. maximum pay gap rules (i.e., if it would be more efficient to have an ideal wage gap that companies should approach or a maximum gap that companies must not exceed, from 0 –*ideal gap* to 100 –*maximum gap*).

A new pre-tested measure of wage-setting criteria (Appendix D) distinguished, also according to an exploratory factor analysis, between criteria associated with low-status workers (i.e., physical exhaustion and shift work, Spearman-Brown = .73) and criteria associated with CEOs (competence, responsibility, leadership ability, organizational ability, and knowledge of languages; $\alpha$ = .78). Participants were asked to rate how much each criterion was important in setting the wage of a person (from 1 –*not at all* to 7 –*extremely*).

Finally, participants were presented with a manipulation check and demographic items (gender, age, education level, work status, general, social, and economic political orientation, SSES, and annual family income), and they were debriefed.

## Results

**Pay gap ratios.** First, we ran a non-parametric Mann-Whitney U test on PGR. As predicted, participants in the highest acceptable condition indicated higher pay gaps than those in the ideal PGR condition, $U = 2240$, $p = .026$, $\eta^2 = .03$. For the linear approach, after excluding outliers (*N* = 6, four men and two women, all in the highest acceptable condition) and log-transforming data, the difference between the two conditions failed to reach significance, $t(143) = -1.55$, $p = .124$, $d = .26$. This may be due to outliers all being in the highest acceptable condition, or because providing the figure led to drastically reduced thresholds of inequality. Still, in line with our hypotheses, PGR was larger for highest acceptable than for ideal PGR (Fig 6). Although the partial divergence between robust and linear models warrants additional studies, the two studies together provide first evidence for our hypothesis that ideal and highest acceptable PGRs are two separate concepts.

In line with the preregistration, we also tested gender differences in pay gaps, through a 2 (condition) x 2 (gender) ANOVA. Please note that we were forced to exclude non-binary

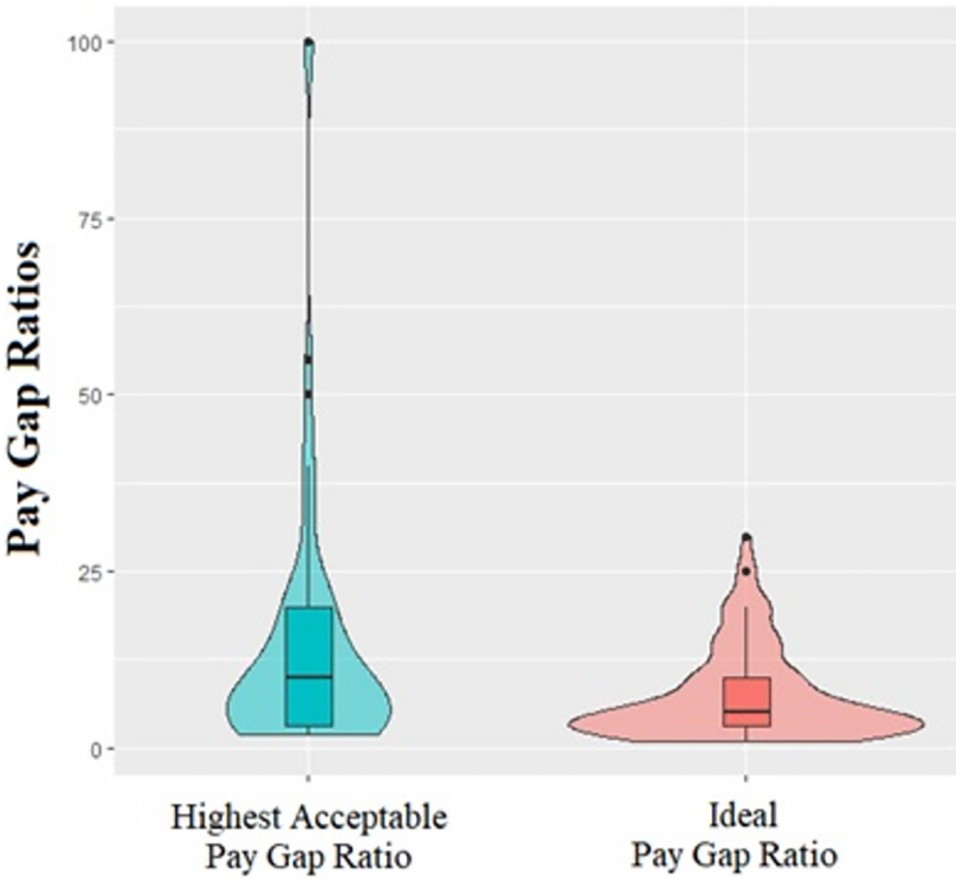

**Fig 6. Ideal and highest acceptable PGRs (Study 3).**

participants due to low sample size. A main effect of gender emerged, $F(1, 139) = 6.44$, $p = .012$, $\eta^2_p = .04$, so that men indicated higher PGRs than women, regardless of type of PGR.

As for motivations, there were no differences between conditions for the items targeted at ideal PGR, $t(148) = 1.59$, $p = .114$, and at the highest acceptable PGR, $t(148) = .12$, $p = .907$, and for the bipolar item, $t(148) = .51$, $p = .609$. Nevertheless, only ideal PGR correlated negatively with equity motivation and positively with preference for merit over equality (see Table 1) suggesting that moral principles may be more relevant for ideal pay-gaps than for maximum acceptable thresholds, as shown in Table 1. When including all motivations as

**Table 1. Correlations between PGR and motivations by type of PGR (Study 2).**

|  | Ideal PGR ($N = 81$) | Highest Acceptable PGR ($N = 64$) |
|---|---|---|
| 2. Equity | -.24* | -.13 |
| 3. Preference for merit | .38*** | .15 |
| 4. Balance between categories | -.16 | -.21 |
| 5. No justifications for a larger gap | -.11 | .02 |

Notes
* $p < .05$
*** $p < .001$.

predictors in a regression, there was no effect for highest acceptable PGR, whereas for ideal PGR preference for merit was the only predictor, $B = 0.22$, 95% CI [0.07, 0.37], $t = 2.87$, $p = .005$.

Therefore, although participants indicated that similar motivations led them to select both PGRs, only ideal PGRs increased with preference for merit.

**Wage-setting criteria.** To test the prediction that the more importance attributed to worker criteria, the lower the PGR would be, we ran a regression model with PGR as dependent variable and CEO and worker wage-setting criteria, condition, and the interaction between the two as predictors. The importance attributed to wage criteria typical of workers negatively predicted PGR, $B = -.19$, 95% CI [-.34, -.04], $t = -2.52$, $p = .013$, while CEO criteria predicted PGR positively, $B = .44$, 95% CI [.25, .64], $t = 4.51$, $p < .001$. Therefore, the less importance was attributed to dimensions typical of the worker's activity, and the more importance was attributed to those typical of the CEO, the higher PGRs were. There was no effect of the two interactions between condition and criteria, suggesting that the criteria played a similar role in establishing ideal and maximum acceptable PGRs.

**Need for legislation and usefulness of ideal vs. Maximum pay gap rules.** Though not preregistered, responses to these two items were skewed towards the two extremes and were thus dichotomized. Regardless of condition, the majority of participants (77%) desired a law that caps executive salaries (i.e., indicated scores higher than the mid-point), binomial $< .001$. Consistently, and again regardless of condition, the majority of participants (62%) deemed it more useful to reduce inequalities by defining a maximum gap that companies must not exceed than to define an ideal wage gap that companies should comply to as best as possible, binomial $< .001$. Those who had such preference were more left-wing ($M = 26.92$, $SD = 18.65$) than those who did not ($M = 37.14$, $SD = 17.10$), $t(146) = 3.34$, $p = .001$. These preferences did not affect PGR, either by themselves or in interaction with condition, all $p$'s $\geq .088$. After excluding outliers, those who believed that an upper limit for the wage gap would be more useful, $t(142) = 1.98$, $p = .050$, indicated smaller gaps for highest acceptable PGR, regardless of condition, all $p$'s $\geq .14$. There was no difference for need for legislation, $t(140) = 1.02$, $p = .310$.

## Discussion

Study 3 provides evidence for our hypothesis that ideal and highest acceptable PGRs are two separate concepts and exist separately in participants' minds, even when investigated in a between-participants design. Nevertheless, the effects of condition were small, which suggests that the comparison between the two values (Study 1) leads participants to magnify their difference. Importance attributed to CEO criteria, on the other hand, predicted higher PGRs. Furthermore, participants deemed a law setting a limit to wage gaps, rather than an ideal gap, as more useful. This may be due to the fact that in this study, the item specified to think about "practical purposes, in order to curb the increasing economic inequalities".

Although participants provided different ideal vs. highest acceptable PGRs, they failed to mention distinct motivations for their decisions. This finding is open to different interpretations. First, participants may indeed have employed the same reasoning to decide both PGRs. Second, they may lack the introspection to understand the reasons that led up to their decisions. Third, the wording of the questions may have been too complex (which could also explain the low correlations for the three pairs that we excluded from the analyses). Further studies should investigate the challenging question of what cognitive and motivational factors lead to defining ideal versus maximum acceptable wage gaps.

## General discussion

Economic theories investigating the effects of wage dispersion have taken two opposing stances, namely the idea that wage inequality enhances ambition and thus motivation, and that it leads to perceptions of unfairness and thus decreases motivation. As of today, economic literature provides evidence for both, as competing and complementary phenomena [43]. Extending this debate on individual perceptions and generic preferences about wage inequality, our studies confirm our hypothesis that individuals have specific thresholds of inequality acceptance, that are affected by individual characteristics such as gender or system-justifying beliefs.

Our studies provide first evidence for our hypothesis that individuals have specific thresholds of inequality acceptance. When asked both PGRs, the large majority of participants indicated thresholds for inequality that exceeded their ideal preference, suggesting that they are able to distinguish the two concepts. This was to some degree also true in Study 3 using a between-participants design. However, the high correlation between PGRs in Study 1 (Appendix G) and the fact that they share some of the predictors (in particular, the endorsement of wage criteria) in Study 3 suggests a certain overlap between the two constructs. At the same time, they are distinct in two ways: first, ideal PGR appears to be relatively homogeneous across participants, whereas highest acceptable PGR has much greater variability. Thus, there seems to be a collective idea of what PGRs should look like ideally, possibly driven by culturally shared principles such as preferences for equality and merit. In line with this interpretation, the second difference regards the distinct predictors. In Study 2, ideal, but not maximum acceptable pay-gaps were predicted by merit beliefs and, to a lesser extent, by concern for equity and fairness. Thus, ideal PGRs seem to be guided by moral considerations, whereas the level of inequality acceptance may be driven by different processes. For example, Plato's main concern was the prevention of negative consequences of inequality on society. Legal limits to inequality (in particular, the 1:4 rule) were intended to prevent "quarrels of long standing" and "disputes among citizens". By preventing the accumulation of wealth, the legislator was expected to contribute to the "happiness" of its citizens and to create a state "free of enmity". Thus, at the center of his arguments were the consequences, not the moral bases, of distribution. Whether this is true also for modern lay reasoning remains a question to be explored in future research.

Three limits of our research should be acknowledged. First, about half of our participants were students, who tend to be younger, more privileged, and better educated than the general population [44]—and social sciences students in particular tend to be more progressive [45]; even their in-lab behavior is substantially different from representative samples [46]. Possibly, a truly representative sample may show larger ranges of inequality acceptance. Second, even though participants were asked to indicate wages for CEOs while keeping in mind the wage of the worker, as of now we cannot prove with certainty that people considered the wage of the worker and adjusted the wage of the CEO accordingly. Further studies need to address this limit more systematically, for example by employing different anchors as worker wages. Third, a number of participants across the first two studies provided higher ideal than highest acceptable PGRs. We are unaware of what considerations may have driven participants to provide such responses, and we cannot exclude the possibility of this being an indicator of low attention, or participants not understanding the task. While analyses ran after exclusions of said participants are largely consistent with the ones reported here, future studies should understand what drives these apparently contradictory responses and possibly include multiple attention or comprehension checks.

## Gender differences

The second main finding of our studies is that PGRs are strongly affected by gender: men feel the need for, prefer, and are willing to tolerate higher levels of wage inequality than women. Interestingly, they also outnumber women among outliers (28 men vs. 11 women, despite the fact that men are only 43% of the total sample across studies). Thus, men are particularly over-represented among those supporting extreme PGRs, which is in line with our finding that gender differences are most pronounced when defining ethically acceptable limits (rather than ideal or lowest acceptable PGRs). Compared to women, men also tend to disregard wage-setting criteria typical of workers and are less likely to support policies that put a cap on executive remuneration. Together, our findings show, consistently across studies and across measures, that men have a much greater tolerance for wage inequality and find higher pay gaps ethically acceptable.

In Study 2, this effect was partially mediated by social dominance orientation: men showed a preference for hierarchically structured society with clear delimitation between social groups–which is consistent with the literature [42]. Attributing higher pays to the CEO may be one way to reinforce the organizational hierarchy. Furthermore, the link between gender and ideal PGR was partially mediated by the importance attributed to aspects of work typically associated with low-status workers. Due to job segregation, women have a higher likelihood of working in low-status jobs than men do and as such may value such type of work more than men do, especially since women are aware that, because of their gender, their own work is usually undervalued [47]. Therefore, valuing low-status work more may lead them to perceive smaller differences between the two categories than men do, and adjust the ideal gaps accordingly.

Nevertheless, SDO and relevance attributed to criteria for workers accounted for only a small part of the gender effect, hence additional variables are likely to be operating. Here we will suggest a number of variables that may contribute to the gender differences observed in our studies. First is self-interest: compared to women, men may feel they have greater chances of reaching and are more entitled to high-status occupations, and as such may indicate greater wages [48]. Women, in contrast, are often barred by the glass ceiling–that is, they do not have access to higher-status positions within companies [49] and may identify less with higher-status referents [36]. Therefore, unfairness perceptions of their lower-status condition may lead women to want to reduce the gap. Future studies may test this, for example, by including a measure of perceived individual mobility instead of general perceptions of social mobility. Secondly, and linked to the above, women may be more inequality-averse than men due to them usually being victims of inequality, both from a social and economic standpoint [50]. This difference becomes even more pronounced for equity-based awards that are often a considerable portion of executive remuneration [51]. Our third suggestion is instead based on a cognitive, rather than a motivational explanation. Previous research has argued that when estimating economic and wage inequality, individuals are strongly anchored to their own perceptions and experiences [21]. Since women are exposed to lower wages than men due to social comparisons, occupational segregation and homophily within their social networks [47, 52], they may be cognitively anchored to lower wages than men and use smaller numbers than men as a reference point. Finally, we cannot exclude the role of ingroup bias, since we only considered male CEOs; future studies may manipulate the gender of the CEO to test whether women would indicate higher wages (and as such higher PGRs) for female CEOs.

## Wage-setting criteria

A third general conclusion that can be drawn from our studies is that ideal and accepted levels of wage inequality are closely intertwined with importance attributed to criteria typically

associated with the CEO and the worker characteristics; in this sense, the roles seem more relevant than the criteria per se. The more participants valued wage criteria typical of workers (such as physical exhaustion), the lower the wage gap they tolerated or found ideal. Thus, our research contributes to the under-investigated question of how wages should be determined and how this relates to people's tolerance for pay gaps. However, given the correlational nature of our studies, we cannot draw any conclusions about the causal link between the two. Our primary interpretation was that subjective wage-setting criteria make people more, or less, tolerant of pay gaps. Still, it is also possible that subjective wage-setting criteria serve as post-hoc justification of one's belief on what pay gaps are acceptable. For instance, if a 1:300 pay gap is considered ethically acceptable, then this belief can be justified on the ground that leadership ability and responsibility are much more important wage-setting criteria than fatigue or shifts. Disentangling the causal relation between the two remains a challenge for future research.

## Policy implications

Although the above studies need to be replicated with representative and possibly cross-national samples, we believe that our findings have important policy implications. While most European countries have minimum wage regulations, only few regulate maximum salaries and generally only for the public and semi-public sector: for example, Italy has introduced a maximum yearly gross salary of €240.000 for public administration personnel in 2014 (art. 13 of the Legislative Decree N. 66 of 24 April 2014). In our (non-representative) sample, the majority of participants supported this kind of regulation, in line with Plato's claim that "the legislator should determine what is to be the limit of poverty or wealth", to ensure that there be "neither extreme poverty, nor, again, excess of wealth" (p. 112). Given that pay gap regulations contain ethical, in addition to economic, considerations, policy makers may be well advised to give greater consideration to public opinion, rather than to rely exclusively on expert advice from economists. Understanding where and why people perceive the ethical limit of pay gaps is an important first step in this direction.

An additional implication derives from the strong gender gap in defining ethically acceptable pay gaps and caps on executive income. Legal constraints on remuneration are decided by parliaments which, however, tend to be composed primarily of men in European countries and in North America. Parliaments with at least 40% women can mainly be found in Africa or Latin America, and only few European nations meet this standard (Sweden, Finland, Norway and Spain; Inter-Parliamentary Union, n.d.). Based on our findings it is plausible to hypothesize that the political empowerment of women (see Global Gender Gap Index) is, if not a prerequisite, at least a predictor of wage gap regulations.

## Conclusion

Previous research on pay gaps has focused on what individuals believe is an ideal level of inequality; this set of studies instead sheds light on a new dimension of attitudes towards economic inequality, namely the threshold of inequality acceptance, and highlights how women have lower such thresholds compared to men. Exploring this threshold and the cognitive and motivational factors that shape it is fundamental to our understanding of attitudes towards economic inequalities.

## Acknowledgments

We are grateful to Gerardo Cervone, who we consulted for identifying items used in the pretest of Study 2 and to Lucia Ronconi for her valuable insight on analyses of Study 2.

## Author Contributions

**Conceptualization:** Carmen Cervone, Andrea Scatolon, Michela Lenzi, Anne Maass.

**Data curation:** Carmen Cervone.

**Formal analysis:** Carmen Cervone.

**Funding acquisition:** Anne Maass.

**Investigation:** Carmen Cervone, Andrea Scatolon, Anne Maass.

**Methodology:** Carmen Cervone, Andrea Scatolon, Michela Lenzi, Anne Maass.

**Project administration:** Carmen Cervone, Michela Lenzi, Anne Maass.

**Supervision:** Michela Lenzi, Anne Maass.

**Visualization:** Carmen Cervone.

**Writing – original draft:** Carmen Cervone, Anne Maass.

**Writing – review & editing:** Carmen Cervone, Andrea Scatolon, Michela Lenzi, Anne Maass.

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
