## [Decision Letter · Decision Letter 0]

11 Jul 2024

PONE-D-23-31636Setting limits: Ethical thresholds to the CEO-worker pay gapPLOS ONE

Dear Dr. Cervone,

Thank you for submitting your manuscript to PLOS ONE. After careful consideration, we feel that it has merit but does not fully meet PLOS ONE’s publication criteria as it currently stands. Therefore, we invite you to submit a revised version of the manuscript that addresses the points raised during the review process.

We look forward to receiving your revised manuscript.

Kind regards,

Nikolaos Georgantzis, Dr.

Academic Editor

PLOS ONE

“This research was supported by grant PRIN 2017 #2017924L2B of the Italian Ministry of Education, University and Research (MIUR), entitled “The psychology of economic inequality”, to the last author.”

Reviewers' comments:

Reviewer's Responses to Questions

**Comments to the Author**

1. Is the manuscript technically sound, and do the data support the conclusions?

Reviewer #1: Yes

2. Has the statistical analysis been performed appropriately and rigorously? 

Reviewer #1: Yes

3. Have the authors made all data underlying the findings in their manuscript fully available?

Reviewer #1: Yes

4. Is the manuscript presented in an intelligible fashion and written in standard English?

Reviewer #1: Yes

5. Review Comments to the Author

Reviewer #1: The paper was very enjoyable to read and it is great to see experimental studies of this nature being undertaken. The three studies in the paper deal with an important issue which ties in with the debate on CEO-employee pay gap or pay ratio. The paper is very well written and clear. The contribution also is very clear and the authors make a good case about the regulatory implications from the findings. I find little to fault but provide some guidance on how to make the paper clearer.

1) The main focus of the study is not clear. It looks like the findings related to gender differences were a by-product; however, this is the focus of a lot of discussion as well as analyses. The early part of the study talks more about the maximum vs. ideal pay gap. I think the writing has to be tightened from the start to clarify the focus. If gender pay gap is the main focus, then make this known from the start and include more references on this. In fact, this part is more interesting than simply testing whether a maximum pay gap is higher than an ideal one, which seems logical.

2) I am not sure why a handful of participants in the studies chose a maximum PGR that is lower than the ideal PGR. My feeling is that they did not read the questions and hence should be removed from all analyses. I know the authors have removed outliers but these cases should not be included in any of the analyses.

3) The theoretical underpinning of the study needs to be expanded. For example, Fehr &Schmidt (1999) present a theory of fairness in the workplace. This study needs to be included but also a wider theoretical discussion is needed. Another theory that is relevant is tournament theory (Lazear & Rosen, 1981).

4) The development of the hypotheses needs to be expanded in line with what is mentioned in point 3.

5) The second hypothesis relates to whether criteria associated with CEOs but not typical of the worker are more accepting of higher PGRs. This is not fully discussed in study 1. The discussion on p.11 relates to hypothesis 1 and gender differences but it is not clear how preference for competence for example relates to tolerance of higher PGRs. The discussion of perceived and ideal criteria (for a sub-group of participants) does not really address this.

References:

Fehr, E., & Schmidt, K. M. (1999). A Theory of Fairness, Competition, and Cooperation. The Quarterly Journal of Economics, 114(3), 817–868.

Lazear, E.P. & Rosen, S. (1981). Rank-order tournaments as optimum labor contracts. Journal of political Economy, 89(5), 841-864.

6. PLOS authors have the option to publish the peer review history of their article (what does this mean?). If published, this will include your full peer review and any attached files.

Reviewer #1: **Yes: **Salma Ibrahim

---

## [Author Response · Author response to Decision Letter 0]

28 Aug 2024

RESPONSE TO REVIEWER COMMENTS

"The paper was very enjoyable to read and it is great to see experimental studies of this nature being undertaken. The three studies in the paper deal with an important issue which ties in with the debate on CEO-employee pay gap or pay ratio. The paper is very well written and clear. The contribution also is very clear and the authors make a good case about the regulatory implications from the findings. I find little to fault but provide some guidance on how to make the paper clearer."

We are delighted that the reviewer enjoyed our work. We greatly appreciate the reviewer's feedback and we believe that the manuscript has greatly improved as a function of this review process. We hope that we adequately implemented all the reviewer’s comments.

"1) The main focus of the study is not clear. It looks like the findings related to gender differences were a by-product; however, this is the focus of a lot of discussion as well as analyses. The early part of the study talks more about the maximum vs. ideal pay gap. I think the writing has to be tightened from the start to clarify the focus. If gender pay gap is the main focus, then make this known from the start and include more references on this. In fact, this part is more interesting than simply testing whether a maximum pay gap is higher than an ideal one, which seems logical."

As we agree that the gender difference is our most interesting finding, we have now clarified the focus on gender differences. In particular, we streamlined the introduction section on maximum vs. ideal pay gap (p. 4), we moved the sections on gender earlier in the introduction and discussion sections (before, rather than after, the wage setting criteria), and we expanded the introduction section on gender differences (p. 6, changes and additions in bold):

“Gender Differences in Preferences for Pay Ratios

Like other disadvantaged groups, women have often been found to endorse politically progressive values and policies. [...] They also tend to be less prone to competition, attribute less positive outcomes to competition (26) and believe less strongly in gender-specific system justification (27).

When it comes to wage inequality, this is also the case. According to the 2023 Global Gender Gap Report by the World Economic Forum, the gender gap is still a global issue, with women being disadvantaged in economic participation and opportunities compared to men across all reviewed countries. For example, in the United States, women currently earn 99¢ for every 1$ earned by men, even when controlling for qualifications and job characteristics (32). In line with such disadvantage, women tend to indicate lower ideal PGRs, namely they prefer somewhat smaller pay gap ratios, compared to men (33,34). These preferences also appear to have a real-word impact: companies led by female CEOs also tend to be more egalitarian, as the CEO wage (but not the worker wage) tends to be lower (35). Put together, this evidence suggests that women may be less tolerant of CEO-worker pay gaps and, possibly, more favorable of smaller caps on executive remuneration compared to men. At the same time, however, there is also evidence pointing to the fact that men and women deem set wages as similarly fair or unfair (36), which would suggest that gender may not affect concerns about wage equality. Therefore, the second aim of this research was to investigate whether gender differences are also present in wage inequality thresholds.”

We have also specified that gender is one of the primary focuses of the paper at the beginning of the manuscript, on p. 4 (additions in bold): 

“This paper aims to explore this threshold in the context of CEO (Chief Executive Officer)-Worker Pay Gap Ratios (henceforth, “PGRs”), and to investigate whether this threshold is affected by gender, as will be outlined below.”

"2) I am not sure why a handful of participants in the studies chose a maximum PGR that is lower than the ideal PGR. My feeling is that they did not read the questions and hence should be removed from all analyses. I know the authors have removed outliers but these cases should not be included in any of the analyses."

While we agree with the reviewer’s point that it is plausible that these participants did not read or understand the questions, we cannot be certain of this. For example, one participant stated in a comment that in our current world, he would believe there is a certain highest acceptable wage for CEOs; however, they said that in an ideal world in which inequality does not impact people, they would give CEOs more money than their highest acceptable threshold. Secondly, given that our first hypothesis is that highest acceptable PGR > ideal PGR, excluding participants who report the opposite may not represent the most rigorous approach and may figure as a questionable research practice. For these reasons, we are reluctant to report the analyses with these exclusions in the main text. 

We have, however, included them in Appendix H of the supplementary materials on OSF: results are consistent with the ones reported in the main text (and often, effects are stronger), with the exception of the SDO mediations in Study 2, which become non-significant. We have now reported both facts in the main text (additions in bold):

Study 1 (p. 11): “Ideal PGR ranged from 1 to 150 (M = 6.49; SD = 14.69) and highest acceptable PGR from 1 to 1000 (M = 18.78; SD = 98.08). Most participants (N = 183, 57%) indicated a higher value for highest acceptable than for ideal PGR, while 115 (36%) indicated the same amount for both variables, and only 24 (7%) indicated lower highest acceptable PGRs than ideal ones. Analyses ran after exclusions of the latter group of participants are available in Appendix H of the supplementary materials.”

Study 2 (p. 16): “Again, most participants (N = 168, 87%) indicated higher values for highest acceptable PGR than ideal PGR, while 22 (12%) indicated equal values, and two indicated lower values. Similarly, most participants (N = 162, 84%) indicated higher values for ideal PGR than for lowest acceptable PGR, whereas 21 (11%) indicated equal values, and 9 (5%) indicated lower values. Finally, only one participant indicated lower values for highest acceptable than lowest acceptable PGR and three participants indicated equal values, whereas the rest of the sample indicated higher values. Analyses ran after exclusions of participants who indicated ideal > highest acceptable PGRs, lowest acceptable > ideal PGRs, and lowest acceptable > highest acceptable PGRs are available in Appendix H of the supplementary materials.”

Study 2 (p. 18): “As for SDO (Fig 4), results of the mediation analyses showed a partial mediation effect for ideal PGR (indirect effect: B = -.04, BootLLCI = -.10, BootULCI = -.003), highest acceptable PGR (indirect effect: B = -.05, BootLLCI = -.13, BootULCI = -.003) and lowest acceptable PGR (indirect effect: B = -.04, BootLLCI = -.10, BootULCI = -.004). It is relevant to note that after exclusions of participants who indicated ideal > highest acceptable PGRs, lowest acceptable > ideal PGRs, and lowest acceptable > highest acceptable PGRs, the mediations are not significant anymore (see Appendix H).”

We also addressed this as a limitation in the General Discussion (p. 29, changes and additions in bold):

“Three limits of our research should be acknowledged. [...] Third, a number of participants across the first two studies provided higher ideal than highest acceptable PGRs. We are unaware of what considerations may have driven participants to provide such responses, and we cannot exclude the possibility of this being an indicator of low attention, or participants not understanding the task. While analyses ran after exclusions of said participants are largely consistent with the ones reported here, future studies should understand what drives these apparently contradictory responses and possibly include multiple attention or comprehension checks.”

"3) The theoretical underpinning of the study needs to be expanded. For example, Fehr &Schmidt (1999) present a theory of fairness in the workplace. This study needs to be included but also a wider theoretical discussion is needed. Another theory that is relevant is tournament theory (Lazear & Rosen, 1981)."

We thank the reviewer for this point. We have now presented the fairness vs. tournament theory debate in more detail and how this relates to our work in the introduction: we tried to keep it brief to comply with point 1 (p. 3, changes and additions in bold):

“Research has shown that a sense of fairness is linked to the willingness to reduce inequality (2), and that the belief that the economic system is illegitimate (hence, unfair) leads to higher demand for the redistribution of resources (3). Whether inequality is considered an incentive or unfair is particularly relevant when discussing CEO (Chief Executive Officer)-Worker Pay Gap Ratios (henceforth, “PGRs”), as this debate holds tangible consequences on organizational policies and individual wages. Organizational models rely on two competing theories, i.e., the tournament theory (4) and the equity and fairness theory (5,6). The tournament model posits that high wage dispersion leads to greater efficiency and productivity in organizations, as workers are motivated to reach higher positions (7). Consistent with this theory, greater differences in pay were shown to be positively linked to performance (8,9); note however that the evidence is mixed (10). On the contrary, the equity model states that high wage inequality leads to decreased productivity, as workers perceive the organization as unfair and are less motivated to make an effort. Consistent with this notion, greater pay differences are negatively correlated with employees’ negative perception of the company (11), diminished perception of leaders’ efficacy and charisma (12), lower integration and poorer performance of top executives (13), and higher likelihood of fraud (14), as well as reduced purchasing intention (15) and a negative opinion of the company (11) by consumers.

The tournament theory and the equity and fairness theory have often been juxtaposed by organizational research, but, to our knowledge, rarely combined. We suggest, instead, that the coexistence of these two notions may be explained by the belief that inequality is good, but only as long as it falls within a certain range.”

"4) The development of the hypotheses needs to be expanded in line with what is mentioned in point 3."

We have now revised the hypotheses section accordingly (p. 8, changes and additions in bold):

“Previous literature has evidenced how wage inequality can simultaneously have positive and negative outcomes on organizations, as a high inequality can provide an incentive for productivity, but also negatively affect employees, management, and the public image of the organization. We argue here that this apparent contradiction is justified by the idea that inequality is preferable – but only within a certain, individual, threshold. Therefore, the main aim of this research is to investigate whether people perceive a threshold of wage inequality that should not be crossed, since it creates inequalities that can not be accepted (regardless the specific features of the work positions). To do so, we asked participants how much a CEO should earn ideally and at maximum, based on the wage of the lowest-paid worker in the same company. We predicted that the highest acceptable wages indicated by participants would exceed ideal ones (Hp1).

Furthermore, given that women (a) hold more egalitarian values, and (b) prefer lower income inequality in societies compared to men, we expected women to desire lower PGRs (Hp 2a). In Study 2 we also tested whether these gender differences be mediated by the differential endorsement of inequality-related ideologies such as SDO, social mobility beliefs and merit beliefs (Hp 2b).

Finally, to explore potential mechanisms underlying ideal and highest acceptable PGRs, the second aim was to investigate the criteria that, in participants’ minds, guide wage setting. We predicted that people who primarily value those criteria that are typically associated with CEOs (e.g., competence), but not those typical of the worker (e.g., fatigue) would also accept and desire higher PGRs (Hp3).”

"5) The second hypothesis relates to whether criteria associated with CEOs but not typical of the worker are more accepting of higher PGRs. This is not fully discussed in study 1. The discussion on p.11 relates to hypothesis 1 and gender differences but it is not clear how preference for competence for example relates to tolerance of higher PGRs. The discussion of perceived and ideal criteria (for a sub-group of participants) does not really address this."

We thank the reviewer for pointing this out. We have now added our considerations on the results concerning criteria (p. 12, additions in bold):

“As for wage setting criteria, our results showed that while participants distinguished between perceived and ideal criteria, criteria themselves did not determine PGRs. One reason for this may be that in Study 1, criteria were not pretested: therefore, our a-priori selection of criteria as more relevant for low-wage workers or CEOs may have not matched with participants’ perceptions (e.g., fatigue could be associated with the physical work of factory workers, but also with the long working hours of managers). We solved this issue in Study 2 with a new, pre-tested measure, that allowed us to reliably separate between criteria that are typical of workers vs. CEOs.”

---

## [Decision Letter · Decision Letter 1]

23 Sep 2024

Setting limits: Ethical thresholds to the CEO-worker pay gap

PONE-D-23-31636R1

Dear Dr. Cervone,

We’re pleased to inform you that your manuscript has been judged scientifically suitable for publication and will be formally accepted for publication once it meets all outstanding technical requirements.

Kind regards,

Nikolaos Georgantzis, Dr.

Academic Editor

PLOS ONE

Additional Editor Comments (optional):

Reviewers' comments:

Reviewer's Responses to Questions

**Comments to the Author**

1. If the authors have adequately addressed your comments raised in a previous round of review and you feel that this manuscript is now acceptable for publication, you may indicate that here to bypass the “Comments to the Author” section, enter your conflict of interest statement in the “Confidential to Editor” section, and submit your "Accept" recommendation.

Reviewer #1: All comments have been addressed

2. Is the manuscript technically sound, and do the data support the conclusions?

Reviewer #1: Yes

3. Has the statistical analysis been performed appropriately and rigorously? 

Reviewer #1: Yes

4. Have the authors made all data underlying the findings in their manuscript fully available?

Reviewer #1: Yes

5. Is the manuscript presented in an intelligible fashion and written in standard English?

Reviewer #1: Yes

6. Review Comments to the Author

Reviewer #1: Thank you for addressing all comments. I am satisfied that they have been addressed appropriately. I believe it is in a publishable state.

7. PLOS authors have the option to publish the peer review history of their article (what does this mean?). If published, this will include your full peer review and any attached files.

Reviewer #1: **Yes: **Salma Ibrahim

---

## [Editor Report · Acceptance letter]

25 Sep 2024

PONE-D-23-31636R1 

PLOS ONE

Dear Dr. Cervone, 

I'm pleased to inform you that your manuscript has been deemed suitable for publication in PLOS ONE. Congratulations! Your manuscript is now being handed over to our production team.

Kind regards, 

on behalf of

Prof. Nikolaos Georgantzis 

Academic Editor

PLOS ONE